# In-House Implementation of Tumor Mutational Burden Testing to Predict Durable Clinical Benefit in Non-Small Cell Lung Cancer and Melanoma Patients

**DOI:** 10.3390/cancers11091271

**Published:** 2019-08-29

**Authors:** Simon Heeke, Jonathan Benzaquen, Elodie Long-Mira, Benoit Audelan, Virginie Lespinet, Olivier Bordone, Salomé Lalvée, Katia Zahaf, Michel Poudenx, Olivier Humbert, Henri Montaudié, Pierre-Michel Dugourd, Madleen Chassang, Thierry Passeron, Hervé Delingette, Charles-Hugo Marquette, Véronique Hofman, Albrecht Stenzinger, Marius Ilié, Paul Hofman

**Affiliations:** 1Université Côte d’Azur, 06000 Nice, France; 2Team 4, Institute for Research on Cancer and Aging Nice (IRCAN), Institut de la Santé et de le Recherche Médicale (INSERM) U1081/CNRS 7284, 06107 Nice, France; 3Laboratory of Clinical and Experimental Pathology, Biobank BB-0033-00025, Centre Hospitalier Universitaire de Nice, 06000 Nice, France; 4FHU OncoAge, Pasteur Hospital, 06000 Nice, France; 5Department of Pulmonology and Thoracic Oncology, Centre Hospitalier Universitaire de Nice, 06000 Nice, France; 6Epione Team, Inria, Sophia Antipolis, 06902 Valbonne, France; 7Department of Oncology, Antoine Lacassagne Comprehensive Cancer Center, 06200 Nice, France; 8Department of Nuclear Medicine, Antoine Lacassagne Comprehensive Cancer Center, 06200 Nice, France; 9Department of Dermatology, Archet II Hospital, Centre Hospitalier Universitaire de Nice, 06000 Nice, France; 10Department of Radiology, Archet 2 Hospital, Centre Hospitalier Universitaire de Nice, 06000 Nice, France; 11Centre Méditerranéen de Médecine Moléculaire (C3M), Institut de la Santé et de le Recherche Médicale (INSERM) U1065, 06204 Nice, France; 12Institute of Pathology, University Hospital Heidelberg, 69120 Heidelberg, Germany; 13Center for Personalized Oncology (DKFZ-HIPO), Deutsches Krebsforschungszentrum (DKFZ), 69120 Heidelberg, Germany

**Keywords:** tumor mutational burden, FoundationOne assay, Oncomine TML assay, lung cancer, melanoma, immunotherapy

## Abstract

Tumor mutational burden (TMB) has emerged as an important potential biomarker for prediction of response to immune-checkpoint inhibitors (ICIs), notably in non-small cell lung cancer (NSCLC). However, its in-house assessment in routine clinical practice is currently challenging and validation is urgently needed. We have analyzed sixty NSCLC and thirty-six melanoma patients with ICI treatment, using the FoundationOne test (FO) in addition to in-house testing using the Oncomine TML (OTML) panel and evaluated the durable clinical benefit (DCB), defined by >6 months without progressive disease. Comparison of TMB values obtained by both tests demonstrated a high correlation in NSCLC (*R*^2^ = 0.73) and melanoma (*R*^2^ = 0.94). The association of TMB with DCB was comparable between OTML (area-under the curve (AUC) = 0.67) and FO (AUC = 0.71) in NSCLC. Median TMB was higher in the DCB cohort and progression-free survival (PFS) was prolonged in patients with high TMB (OTML HR = 0.35; FO HR = 0.45). In contrast, we detected no differences in PFS and median TMB in our melanoma cohort. Combining TMB with PD-L1 and CD8-expression by immunohistochemistry improved the predictive value. We conclude that in our cohort both approaches are equally able to assess TMB and to predict DCB in NSCLC.

## 1. Introduction

Immune checkpoint inhibition (ICI) has dramatically revolutionized treatment in various cancers [1], notably in non-small cell lung cancer (NSCLC) and melanoma [2,3]. Since not all patients respond equally to ICI, robust predictive biomarkers are urgently needed to appropriately select patients. Until now, the expression of Programmed Death-Ligand 1 (PD-L1) as assessed by immunohistochemistry (IHC) in tumor cells has been the only FDA-approved biomarker for the selection of patients undergoing ICI in NSCLC [4]. However, PD-L1 IHC has limitations in the prediction of durable clinical benefit (DCB) in patients treated by ICI [5,6]. Additionally, PD-L1 expression in tumor-infiltrating immune cells as well as the tumor infiltration of CD8^+^-lymphocytes has been studied as potential biomarkers in ICI [7,8,9,10]. Likewise, tumor mutational burden (TMB), defined as number of mutations per megabase of exonic DNA, has been proposed as a novel promising biomarker for the prediction of response to ICI and has been validated across different cancer entities [11,12,13,14,15], especially in NSCLC [16,17]. Initially, TMB was evaluated using whole exome sequencing (WES) [12,18,19], but for implementation in routine clinical practice, specific targeted sequencing panels have been developed [14,20,21]. The data obtained from these targeted sequencing panels showed significant correlations with WES datasets and predictive power across several solid tumor types [19,22,23]. However, the panel size must meet specific requirements to allow the precise detection of TMB [24,25].

While some “in-house” solutions are available at specific institutions such as the MSK-IMPACT panel at the Memorial Sloan Kettering Cancer Center, others require samples to be sent to certified testing providers for analysis, like the FDA-approved FoundationOne^®^ (FO) test. So far, routine assessment of TMB using commercially-available panels for “in-house” use has been notably absent. In this context, robust determination of TMB needs to be standardized to allow its broad implementation in routine clinical practice [24,26,27].

The purpose of this study was to analyze the novel amplicon-based Oncomine™ TML (OTML) panel [21], having a genomic footprint spanning over 1.65 Mb of DNA across 409 oncogenes, to assess TMB in a cohort of 60 and 36 NSCLC and melanoma patients, respectively, and to subsequently predict DCB to ICI treatments. We have compared these results to those from the FO assay to directly assess the correlation between two commercially available targeted sequencing panels. Additionally, we assessed PD-L1 expression in tumor cells (TC) and tumor-infiltrating immune cells (IC), as well as CD8^+^-lymphocyte infiltration by IHC and we created mathematical models to integrate all these different biomarkers into a multivariate analysis to define the best set of biomarkers for the prediction of DCB in NSCLC and melanoma patients undergoing ICI.

## 2. Results

### 2.1. Description of Patient Cohort

In total, 96 patients were included in the patient cohort: 60 patients with NSCLC and 36 patients with melanoma (Appendix A). In this cohort, the clinical follow-up for patients under immunotherapy in first- or second line was available for 48/60 (80%) patients in the NSCLC cohort. Five NSCLC patients ultimately did not receive ICI, while for seven NSCLC patients, no clinical data to compute DCB was available (Appendix A). In the melanoma cohort, clinical data for all patients were available (Appendix A). Patient characteristics are summarized in Appendix A (NSCLC) and Appendix A (melanoma). Additionally, all data are provided in Appendix A. Mutations detected by FO for the 30 most mutated genes are shown in Appendix A. The median turn-around time (TAT) for samples sent to Foundation Medicine was 15 business days (range: 11–40) for NSCLC samples and 13 business days (range: 9–44) for melanoma samples (Appendix A). The implemented workflow using the OTML panel in our laboratory allowed us to obtain final results within 5 working days and within 7 working days when the mutation analysis using the Hotspot Panel v2 was included.

### 2.2. Deamination in FFPE Samples

The OTML panel assesses the level of deamination and provides a deamination score, where a higher value indicates more deamination. Detectable deamination (deamination score > 0) occurred in 65% of all tested samples (Appendix A) and emerged as an important issue during the testing period. As this has had a direct impact on the correlation to the FO results, different cut-off points were assessed (Appendix A). A deamination score of 40 was considered to be a reliable cut-off with a high correlation to FO, allowing the classification of most of the samples as samples with a higher deamination score were filtered out (Appendix A). Deamination repair using an Uracil-DNA glycosylase (UDG) treatment either during extraction or by adding UDG prior to library preparation was compared in a training test of 8 samples (Appendix A). As both protocols were able to reduce deamination scores, UDG treatment prior to the library preparation was implemented during the testing period as it demonstrated efficient deamination reduction while being easy to implement, requiring only 30 min in total, and was utilized for 11/60 (18%) lung cancer and 12/36 (33%) melanoma samples.

### 2.3. Tumor Mutational Burden (TMB) Assessed by Targeted Sequencing Panels are Well Correlated

The OTML panel and the FO test report TMB as mutations/megabase of exonic DNA extrapolated from the sequenced genes, but FoundationOne^®^ also includes synonymous mutations. While the two TMB values are well correlated for lung cancer samples (*R*^2^ = 0.73), the correlation is even better for the melanoma samples (*R*^2^ = 0.94, Figure 1A,B, left panels). In both cases, samples with a percentage of tumor cells greater than 50% in tissue section were better correlated (NSCLC *R*^2^ = 0.78, melanoma *R*^2^ = 0.96) than samples below that threshold (NSCLC *R*^2^ = 0.47, melanoma *R*^2^ = 0.79, Figure 1A,B, right panels). However, more samples with lower tumor cell content were successfully sequenced using the OTML panel than with FO (Appendix A) and the ratio of successfully sequenced samples was greater in melanoma samples than in lung cancer samples (Appendix A).

Since the OTML panel was not validated to report specific mutations but only TMB, we used the Hotspot V2 panel in parallel which is able to detect hotspot mutations in 50 selected genes. Indeed, the detection of mutations using the FO panel was well correlated with the Hotspot V2 panel (Appendix A).

### 2.4. TMB Assessed by Targeted Sequencing Panels in the Same Patients is Associated with DCB in NSCLC but not in Melanoma

The median TMB was higher in the group with DCB in the samples tested with FO (median TMB for DCB = 17 mutations/Mb, NDB = 11 mutations/Mb; Mann–Whitney *p* = 0.0477) and the OTML panel (median DCB = 9.39 mutations/mb, NDB = 5.88 mutations/Mb, Mann–Whitney *p* = 0.0884) without reaching statistical significance in the OTML panel (Figure 2A). ROC curves were computed on the whole available dataset for both TMB assessed by FO and using the OTML panel (Figure 2B). While the FO test generated a greater area under the curve (0.71 vs. 0.67) than the OTML panel, the results changed when computing the ROC on the common dataset, where TMB data was present from both panels on the same sample (Figure 2C). Both panels improved with a larger area under the curve (AUC) in the OTML panel (AUC = 0.77 vs. 0.77) when selecting the common dataset. Prediction of the DCB using five-fold cross validation was comparable between the two panels (Appendix A).

In the NSCLC cohort, the AUC for TMB was greater than the AUC computed for PD-L1 TC and the CD8 score for the prediction of DCB, irrespective of the panel used (Appendix A) and PD-L1 expression (TC) was independent of TMB (Appendix A). However, the assessment of PD-L1 in IC was associated with DCB (Appendix A).

Optimal cut-off values to define the “high TMB” NSCLC population were computed using the minimum distance method and Youden index and are summarized in Appendix A together with sensitivity and specificity. Both methods defined a TMB of greater or equal to 9.39 mutations/Mb for the OTML and greater or equal to 15 mutations/Mb for the FO test as a “high TMB” population. Indeed, the proportion of patients with DCB was higher in the high TMB cohort defined by FO (RR = 0.54, 95% CI = 0.27–0.94, Fisher Exact test *p* = 0.0255) and using the OTML panel (RR = 0.39, 95% CI = 0.15–1.03, Fisher Exact test *p* = 0.0178; Figure 2D). Likewise, PFS in the high TMB population was longer with the OTML panel (12.1 months vs. 1.7 months, HR = 0.35, 95% CI = 0.17–0.86, *p* = 0.0200) and FO (9.3 months vs. 1.7 months, HR = 0.45, 95% CI = 0.20–0.93, *p* = 0.0319, Figure 3A,B). Overall survival data was not mature enough to be computed at the time of this publication. The clinical follow-up including PD-L1 expression and CD8 expression is shown in Figure 4 for each lung cancer patient individually.

In contrast to NSCLC, TMB was not a predictive biomarker in the melanoma cohort (Figure 5) and median TMB did not differ between the DCB and the NDB population (FO = 17 vs. 18 mutations/Mb, OTML = 10.22 vs. 8.51 mutations/Mb, Figure 5A). There were no significant differences in the high TMB population at the computed cut-offs (FO = 18 mutations/Mb, OTML = 5.06 mutations/Mb, Figure 5B, Appendix A) and PFS was comparable (Figure 6).

### 2.5. Combining TMB with PD-L1 Expression in Immune Cells Can Improve Prediction of DCB in NSCLC

We assessed the predictive power of the combination of TMB with other biomarkers (CD8^+^-score, TC, and/or IC PD-L1 in tumor) using decision tree and logistic regression models (see Appendix A). While the combination of TMB with PD-L1 TC and CD8^+^-score only demonstrated a limited benefit, we saw an improved prediction of DCB when combining TMB with PD-L1 expression in IC using logistic regression models, whereas decision trees were not able to improve the results (Appendix A, Appendix A). Interestingly, combining PD-L1 in IC with TC PD-L1 generated the highest AUC using ROC (Appendix A).

Besides, the packyears of smoking metric was the most predictive biomarker for DCB in NSCLC in our cohort and it improved prediction when combined with the other biomarkers including TMB (Appendix A, Appendix A).

Finally, combining biomarkers in melanoma patients did not improve prediction of DCB in our cohort (Appendix A).

## 3. Discussion

Here, we report the implementation of “in-house” TMB assessment in routine clinical practice using the Oncomine™ TML panel with an Ion Genestudio™ S5. This was compared to the FoundationOne^®^ test that requires samples to be sent to a certified testing provider. This study is the first comparison of two large targeted sequencing panels in a daily practice that can be used for the determination of TMB.

The OTML and the FO tests exhibited excellent correlation in our melanoma cohort but still good concordance in NSCLC, thus confirming previous comparisons of the OTML panel to WES data [21]. However, the correlation was strongly dependent on the percentage of tumor cells in tissue sections [28]. Additionally, deamination that occurs in FFPE samples was a significant issue in our cohort when using the OTML panel. This could be reduced by implementing an enzymatic deamination repair using UDG, confirming previously reported results [29]. For future studies, we would therefore recommend that one generally includes the UDG-treatment when using the OTML panel for TMB assessment to overcome FFPE-related artefacts [30,31]. For this study, we determined a deamination score of 40, as assessed by the OTML panel, to be a reliable cut-off to filter out samples with excessive deamination. However, this value needs further investigation and a standardized cut-off will be fundamental for routine clinical use. Furthermore, the OTML panel is an amplicon-based sequencing panel, while the FO test is based on hybrid-capture enrichment, which might lead to some discordances, as has been previously demonstrated [32]. Likewise, different tumor sections were used for the analysis using FO and OTML, which might have also affected the correlation between the two respective tests due to underlying tumor heterogeneity [33]. Our data demonstrated that, besides some limitations due to low tumor cell ratio and impaired sample quality due to FFPE sequencing artefacts, TMB assessment from FFPE tissue using the OTML panel can be implemented in routine clinical care, which is consistent with previously published results on the same panel [34]. However, it is of the highest importance to master the preanalytical phase for the precise estimation of TMB [24].

In routine clinical use, the TAT to get the sequencing results is critical to be able to start appropriate treatment. Using the OTML panel, we were able to get results from DNA to the final report in five working days, however, this is highly dependent on the number of requested samples to start a sequencing run. In contrast, Foundation Medicine claims to get the final report within 14 days plus shipping time but is independent of the number of samples. In our study this limit was often exceeded, and for some samples the final report was obtained after more than 30 working days, a delay that impeded the implementation of the appropriate treatment in patients.

To demonstrate that TMB can be implemented in routine clinical care, we also associated the TMB data obtained by the two panels to the clinical outcome of the patients. Unfortunately, clinical data was not available for every patient from the NSCLC cohort and failed assays allowed us to directly compare only a subset of 30 NSCLC patients and 29 melanoma patients, where the clinical data as well as the successful sequencing data from both tests were available. Therefore, we also analyzed the panels independently using the maximum amount of information available for each of the respective panels (36 patients for NSCLC and 32 patients for melanoma; Figure 1).

In our NSCLC cohort, both panels were indeed able to predict DCB in patients treated with ICI and the proportion of DCB patients was significantly increased in patients with high TMB (Figure 2). Additionally, PFS was prolonged in the NSCLC patients with high TMB (Figure 3) demonstrating that TMB was indeed a valuable biomarker in NSCLC in our cohort. The cut-off to determine the high TMB population was based on the ROC curves and these were concordant when using the Youden index or minimum-distance methodology. A cut-off of 9.39 mutations/Mb was specified for the OTML panel. However, independent studies are required to evaluate whether this is generally suitable in NSCLC. The calculated cut-off of 15 using the FO test was higher than the cut-off of 10 that was previously reported in CheckMate-227 and CheckMate-568 [16,17]. However, the cut-off of 10 was based on a combinatorial treatment of Nivolumab plus Ipilimumab while the patients in this cohort were treated with Pembrolizumab or Nivolumab as single agent. Consequently, a larger validation study, incorporating different treatments and tumor entities is necessary to define valid cut-offs using the described test.

Most importantly, both panels were comparably able to predict DCB in our NSCLC cohort. Interestingly, the AUC of the ROC was increased in samples where the sequencing run was successfully performed on both tests, presumably due to better quality samples. Nevertheless, we could demonstrate that the cut-off value is dependent on the respective panel and should be calculated individually for each panel and cancer entity based on clinically-validated samples [34,35]. To improve prediction of DCB in our cohort, we included a multivariate analysis including PD-L1 expression from both TC and IC population as well as CD8^+^ lymphocyte infiltration, which yielded only a minor improvement in the prediction performance. However, the size of our cohort might have been too limited to draw universal conclusions from these data.

Unfortunately, we could not confirm promising results on the use of TMB in melanoma samples as both panels failed to predict DCB or PFS differences in this cohort [15,36,37]. Further investigation is therefore critical to improve our understanding of TMB in melanoma. Given that our cohort was rather small and more importantly included very few samples with high TMB, we could speculate that the cohort size was too limited to allow the detection of clinical response based on TMB. However, the cut-off was determined by calculating the minimal distance method and Youden index, which is limited by the low area under the curve and consequently, the calculated cut-offs might not be clinically or statistically meaningful. In this context, it is noteworthy that a recent analysis from a large melanoma patients cohort failed to show a significant correlation between high TMB and survival even at a high TMB cut-off of 30.7 [14]. This highlights the requirement for further research on the role of TMB as a predictive biomarker in melanoma, especially in a real-life setting and stratified for different treatments. However, our study is limited in size and consequently, the treatment outcome was analyzed without stratifying the patients for their respective treatment and by combining both first-line and second-line treatment.

## 4. Materials and Methods

### 4.1. Patient Selection

We consecutively included patients treated by first- or second-line ICIs at the Nice University Hospital and Centre Antoine Lacassagne (Nice, France) whose treatment started between November 2016 and October 2018 with sufficient tumor material to enable all planned analyses. Clinical outcome was assessed by an independent radiologist following RECIST v1.1 [38]. DCB was defined by at least 6 months with no progressive disease. Patients progressing before 6 months were classified as patients with no durable benefit (NDB). Progression-free survival (PFS) was determined by time from treatment initiation to progressive disease or death, whatever occurs first. The study was performed in accordance to the guideline of the declaration of Helsinki, approved by the local ethics committee (CHUN, IE-2017-905) and all patients provided written informed consent.

### 4.2. Sample Preparation

Samples with a content of at least 20% tumor cells were selected for either in-house sequencing using the Oncomine™ TML panel and Ion AmpliSeq Cancer Hotspot Panel v2 (both Thermo Fisher Scientific, Waltham, MA, USA) or sent out for external testing using the FoundationOne^®^ test (Foundation Medicine, Cambridge, MA, USA). A detailed description of the sample preparation and sequencing is outlined in Appendix A. PD-L1 IHC (22C3 PharmDx assay; Dako Agilent, Santa Clara, CA, USA), in addition to CD8^+^ IHC expression analyses (clone SP57; Roche Ventana, Tucson, AZ, USA) were performed using a Ventana BenchMark ULTRA (Roche Ventana), as previously described [39,40] and outlined in Appendix A.

### 4.3. Statistical Analyses

The correlation of TMB between the different sequencing panels was analyzed using coefficient of determination. Differences between categorical variables were assessed using Fisher’s exact test or χ^2^. TMB differences between the groups were assessed using the Mann–Whitney test. The predictive power of TMB was calculated using receiver operator characteristics (ROC) with the computed area under the curve (AUC). To determine the best cut-off, the Youden-index and the closest-proximity to the c(0,1) corner has been utilized. Kaplan–Meier curves for PFS were validated using a log rank test. Logistic regression models and decision trees were built using the Scikit-learn software [41] and are described in Methods S5. A *p*-value < 0.05 was considered to be statistically significant and Bonferroni adjustment of *p*-values was performed where appropriate. All statistical analyses were performed using GraphPad Prism (version 5.0, GraphPad Software Inc., San Diego, CA, USA), R (version 3.5.0, R Foundation for Statistical Computing, Vienna, Austria) or Python (version 3.7.2, Python Software Foundation, Wilmington, DE, USA).

## 5. Conclusions

In summary, while our study is an explanatory analysis, we demonstrated that the FO and the OTML assays can equally be used to assess TMB in a routine clinical setting allowing the in-house assessment of TMB using FFPE samples. However, samples of sufficient quality regarding tumor cell content and deamination should be used for future studies.

TMB is a predictive biomarker of ICI response in NSCLC. However, we failed to confirm promising data on TMB as a predictive biomarker of ICI response in melanoma. Further standardization and multicentric validation are necessary to be able to implement TMB in routine clinical diagnosis using the OTML panel.

## Figures and Tables

**Figure 1 cancers-11-01271-f001:**
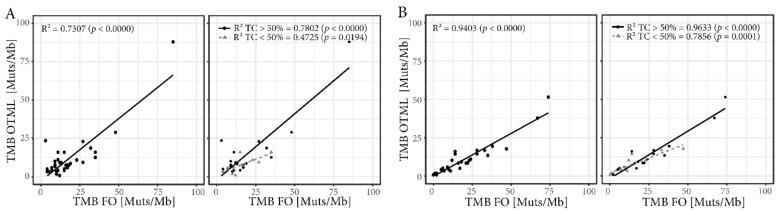
Correlation of tumor mutational burden (TMB) between the targeted sequencing panels. Correlation of TMB assessed by Oncomine TML (OTML) and by FoundationOne (FO) in NSCLC (**A**) and melanoma (**B**) (left panels). The correlation of the TMB is influenced by the percentage of tumor cells (TC) in the tissue sections for the respective samples (right panels) with a greater correlation in samples with a tumor cell ratio higher than 50%.

**Figure 2 cancers-11-01271-f002:**
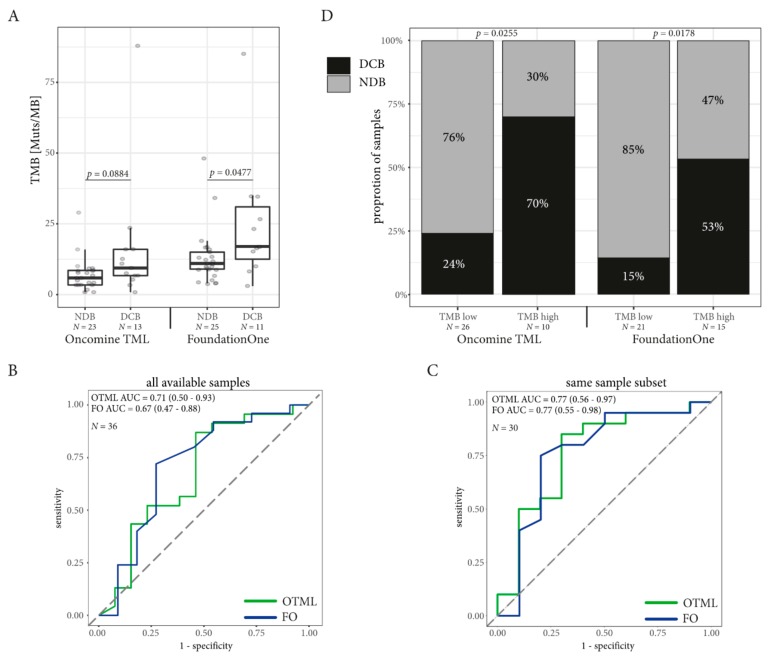
TMB as Biomarker in non-small cell lung cancer. The cut-off for high TMB population for the Oncomine TML panel (OTML) is set to 9.39, while the cut-off for high TMB population in the FoundationOne test (FO) is set to 15 in the NSCLC population. (**A**) The mean TMB is higher in the DCB cohort than in the NDB cohort. (**B**) The ROC curves with their respective areas under the curve (AUC) are computed on all available data for each test individually. The 95% confidence interval is indicated in brackets. (**C**) The ROC curves with their AUC in the subpopulation where the TMB was correctly assessed using both panels in the same sample cohort allowing direct comparison of the different TMB panels. The 95% confidence interval is indicated in brackets. (**D**) The ratio of NSCLC patients with a durable clinical benefit (DCB) is greater than the patients with no durable benefit (NDB) in the cohort with high TMB. The number of samples (*N*) used for calculation is mentioned on each figure.

**Figure 3 cancers-11-01271-f003:**
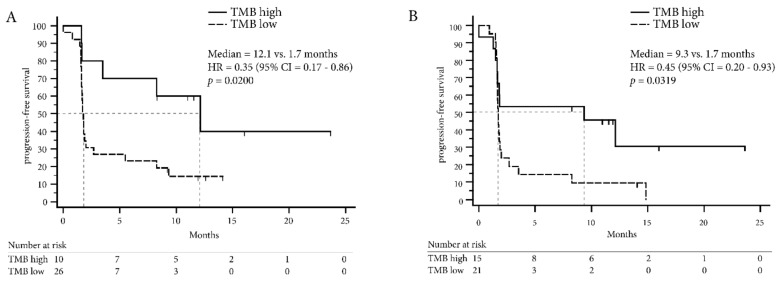
Progression-free survival according to TMB in non-small cell lung cancer patients. (**A**) Progression-free survival (PFS) is computed for the NSCLC dataset using the Oncomine TML panel (OTML) and (**B**) the FoundationOne test (FO).

**Figure 4 cancers-11-01271-f004:**
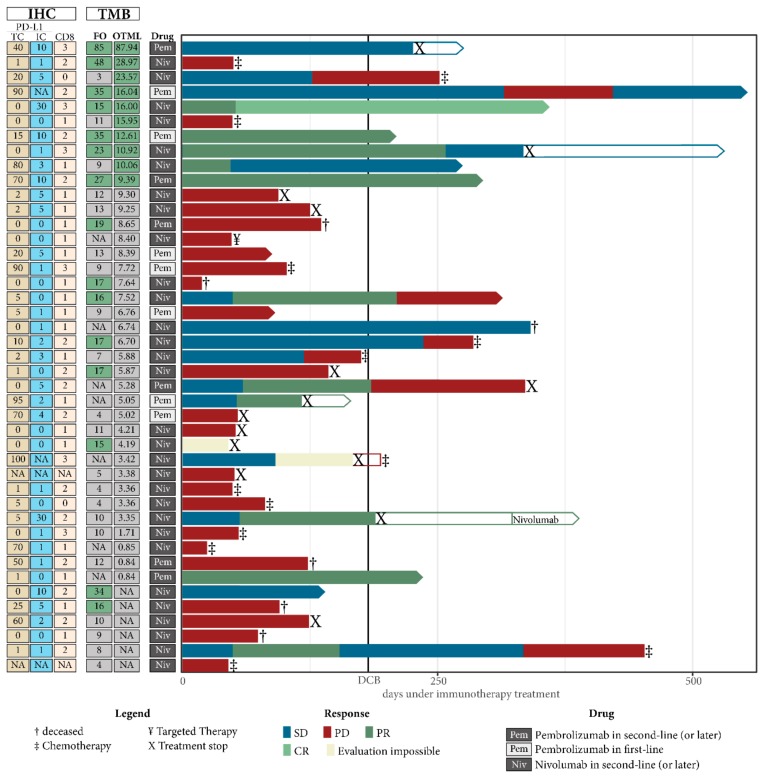
Individual clinical outcome dependent on TMB. Each patient where TMB was successfully obtained using either one of the two panels is represented by one horizontal column. Only patients undergoing immune-checkpoint inhibitor treatment are shown and samples are ordered from highest to lowest TMB as obtained using the Oncomine TML panel (OTML), followed by FoundationOne result (FO). TMB values that were classified to be in the “TMB high” cohort are marked with a green filling. The response according to RECIST v1.1 is color coded on each bar representing the individual follow up of each patient. The respective drug as well as the results obtained by sequencing and immunohistochemistry are shown on the left of the graph.

**Figure 5 cancers-11-01271-f005:**
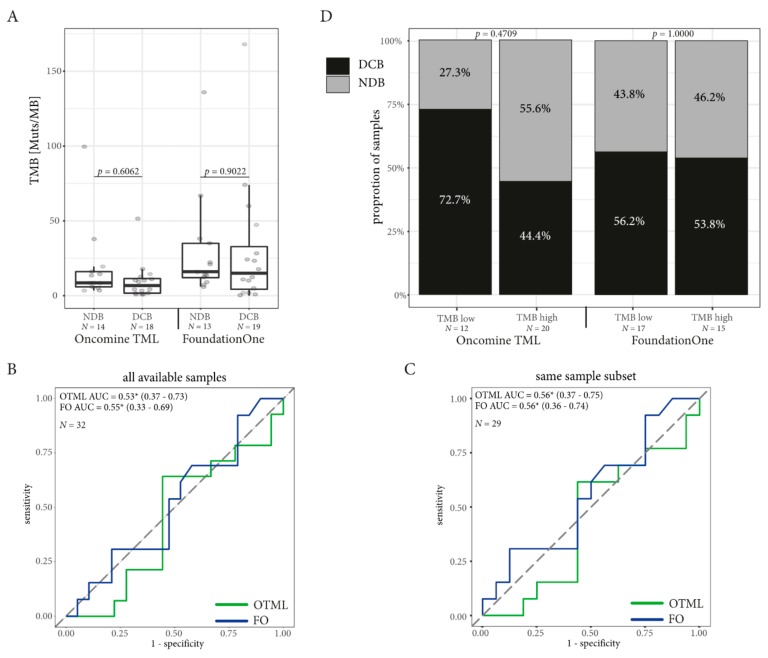
TMB as Biomarker in melanoma. The cut-off for high TMB population for the Oncomine TML panel (OTML) is set to 5.06 while the high TMB population in the FoundationOne test (FO) is defined by a TMB greater than 18 in the melanoma population. (**A**) There is no difference in the mean TMB between the DCB and the NDB cohort in melanoma. (**B**) The ROC curves with their respective areas under the curve (AUC) are computed on all available data for each test individually. The 95% confidence interval is indicated in brackets. The asterisk indicates that the ROC curve favors the low TMB cohort and not the high TMB cohort. (**C**) The ROC curves with their AUC in the subpopulation where the TMB was correctly assessed using both tests in the same sample cohort allowing direct comparison of the different TMB panels. The 95% confidence interval is indicated in brackets. Again, the asterisk indicates that the low TMB cohort is favored in both panels. (**D**) There is no significant difference in the ratio of DCB vs. NDB patients according to TMB independently of the panels used. The number of samples (*N*) used for calculation is mentioned on each figure.

**Figure 6 cancers-11-01271-f006:**
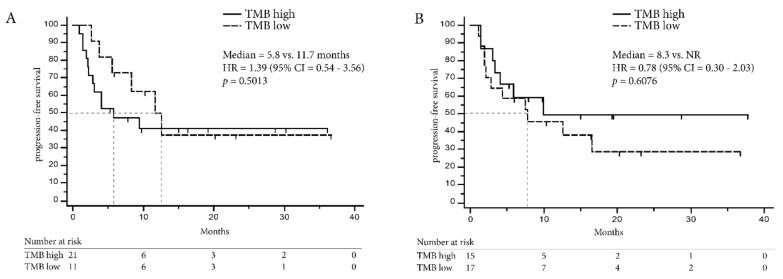
Progression-free survival according to TMB in melanoma patients. (**A**) Progression-free survival (PFS) is computed for the melanoma dataset using OTML and (**B**) FO NR = median PFS was not reached.

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
