# Peer review of "In-house Implementation of Tumor Mutational Burden Testing to Predict Durable Clinical Benefit in Non-small Cell Lung Cancer and Melanoma Patients"

_cancers, 2019, doi:10.3390/cancers11091271_

Round 1
Reviewer 1 Report
This is a good manuscript. I recommend publication after making changes below and re-review.
1) This publication presents detailed granular data in nearly every figure. These data should all be included in the manuscript as supplemental tables as well. A large part of the scientific value of this manuscript is the new data presented, so data tables are required to allow efficient re-analysis of these data. I do not recommend publication of this manuscript without a data table to support nearly every figure.
2) Numerous studies have established that TMB is predictive of immunotherapy benefit in melanoma. The data included in this paper is under-powered to test this hypothesis. The manuscript should be significantly revised to address this fact.
3) The following sentence in the abstract is confusing. , "median TMB was higher in the DCB 42 cohort and PFS was prolonged (OTML HR = 0.35; FO HR = 0.45).” PFS was prolonged in which cohort?
4) The scale for the x-axis of the panels in in figure S4 should agree.
Author Response
This is a good manuscript. I recommend publication after making changes below and re-review.
1) This publication presents detailed granular data in nearly every figure. These data should all be included in the manuscript as supplemental tables as well. A large part of the scientific value of this manuscript is the new data presented, so data tables are required to allow efficient re-analysis of these data. I do not recommend publication of this manuscript without a data table to support nearly every figure.
Reply: We totally agree with the reviewer that it is important to include all data with this publication to allow deeper assessment of the underlying data, especially for laboratories who are conducting or planning to conduct similar studies. We have therefore added the dataset as supplementary data 2.
2) Numerous studies have established that TMB is predictive of immunotherapy benefit in melanoma. The data included in this paper is under-powered to test this hypothesis. The manuscript should be significantly revised to address this fact.
Reply: We agree with the reviewer that the study cohort is too small to draw universal conclusions. We have modified the discussion accordingly to clarify the limitations of the data analysis in the melanoma cohort.
3) The following sentence in the abstract is confusing. "median TMB was higher in the DCB 42 cohort and PFS was prolonged (OTML HR = 0.35; FO HR = 0.45).” PFS was prolonged in which cohort?
Reply: We apologize for the confusion and we have now modified the sentence in the abstract to clarify this statement.
4) The scale for the x-axis of the panels in in figure S4 should agree.
Reply. We have now harmonized the axes of Supplementary Figure 4.
Reviewer 2 Report
In this study, authors clarified the usefulness of the novel amplicon-based Oncomine™ TML (OTML) 73 panel, having a genomic footprint spanning over 1.65 Mb of DNA across 409 oncogenes, to assess 74 TMB in a cohort of 60 and 36 NSCLC and melanoma patients and to subsequently 75 predict DCB to ICI treatments. They conclude that TMB with PD-L1 expression of tumor cells and CD8-expression of TIL approaches are equally able to assess TMB and to predict DCB in NSCLC.
However, there seems to be some major limitations in this paper.
1. There were few studies that verified the concordance rate between TMB assessed with panel sequence and that of with WES, a report comparing TMB assessed with MASK-IMPACT panel with that of with WES (Rizvi H, et al. JCO 2018) and a report of comparison of TMB using panel sequence of cell free DNA with WES result in OAK study (Gandara DR , et al. Ann Oncol 2017).
Each panel sequence is a capture sequence method of Illumina based system using DNA extracted from fresh frozen samples, and subtraction with germ line sequencing results is also performed.
The limitation in this study is that FFPE samples where deamination of nucleotides causes C:G>T:A transition and the subtraction with the results of germ line sequencing results is not performed, and furthermore, ampliSeq that is more susceptoible to PCR efficiency than captured based sequencing method is used.
It should be used to verify the usefulness of TMB as an efficacy prediction tool for ICI treatment after showing the concordance rate with WES, and it is unreasonable to use this approach as a response predictor of ICI in real practice.
2. The number of TMB (/Mbp) is required.
3. There is too much difference in number between lung cancer and malignant melanoma cases. In addition, there are some cases of lung cancer not receiving ICI treatment, and the variation of these cohorts is too large, which seems to be unsuitable for analysis.
Author Response
In this study, authors clarified the usefulness of the novel amplicon-based Oncomine™ TML (OTML) 73 panel, having a genomic footprint spanning over 1.65 Mb of DNA across 409 oncogenes, to assess 74 TMB in a cohort of 60 and 36 NSCLC and melanoma patients and to subsequently 75 predict DCB to ICI treatments. They conclude that TMB with PD-L1 expression of tumor cells and CD8-expression of TIL approaches are equally able to assess TMB and to predict DCB in NSCLC.
However, there seems to be some major limitations in this paper.
There were few studies that verified the concordance rate between TMB assessed with panel sequence and that of with WES, a report comparing TMB assessed with MASK-IMPACT panel with that of with WES (Rizvi H, et al. JCO 2018) and a report of comparison of TMB using panel sequence of cell free DNA with WES result in OAK study (Gandara DR , et al. Ann Oncol 2017).
Each panel sequence is a capture sequence method of Illumina based system using DNA extracted from fresh frozen samples, and subtraction with germ line sequencing results is also performed.
The limitation in this study is that FFPE samples where deamination of nucleotides causes C:G>T:A transition and the subtraction with the results of germ line sequencing results is not performed, and furthermore, ampliSeq that is more susceptoible to PCR efficiency than captured based sequencing method is used.
It should be used to verify the usefulness of TMB as an efficacy prediction tool for ICI treatment after showing the concordance rate with WES, and it is unreasonable to use this approach as a response predictor of ICI in real practice.
Reply: We totally agree with the reviewer that previous studies have exclusively used hybrid-based capture and Illumina sequencing. Therefore, one of the aims of the present study was to investigate whether amplicon-based sequencing and sequencing using IonTorrent technology are also feasible for the assessment of TMB. As seen in Figure 1B, the correlation between the two assays is very good in the melanoma samples (with an R² of 0.94). We would consequently not support the statement that ampliSeq, in general, is an unreasonable approach for the assessment of TMB. Additionally, the data presented in this manuscript demonstrates that TMB assessed using the ampliSeq panel is correlated with clinical response and is able to predict durable clinical benefit in lung cancer patients undergoing anti-cancer immunotherapy.
The lower correlation in lung cancer patients is presumably (and as discussed in the manuscript) due to the underlying sample, for example due to varying tumor cell contents as demonstrated in Figure 1 in the manuscript. This assumption is supported by a recent article by Kazdal D et al. (https://doi.org/10.1016/j.jtho.2019.07.006) demonstrating differences in TMB depending on the sample region due to underlying tumoral heterogeneity.
Consequently, we believe that our study demonstrates that ampliSeq-based sequencing is indeed feasible for the assessment of TMB and that a hybrid-based capture approach is not a requirement. We have consequently clarified this in the discussion to better highlight the differences between the panels and also to discuss limitations to the interpretation of the data.
We also totally agree with the reviewer that FFPE tissue is challenging to use due to the described deamination errors. Therefore, we have investigated this problem and demonstrate (see supplementary figure 5) that the correlation we see between the assays is dependent on the deamination in the sample. However, we also demonstrate that the correlation remains relatively stable when applying a stringent filter based on the deamination assessment provided by the Oncomine Panel. The assessment of deamination in the panel is unique to the Oncomine TML panel and is very useful as it allows filtering of highly deaminated samples thereby avoiding false-high TMB values.
Additionally, we show that the amount of deamination can be substantially reduced using a simple Uracil-DNA glycosylase incubation. This step can be done directly prior to the library preparation or during isolation of DNA with appropriate DNA isolation kits, like the Qiagen GeneReader FFPE kit. Additionally, other vendors sell enzymes or whole mixes to reduce FFPE-related errors, but assessing all of these was outside of the scope of this investigation. Importantly, our results are in line with another publication (Ref 33, Endris V et al, Int J Cancer, 2019) demonstrating that FFPE tissue can be used in routine clinical care. We have added this to the discussion to better demonstrate the limitations.
As mentioned, the most commonly used sequencing tests for the assessment of TMB are the MSK-IMPACT assay, developed by the Memorial Sloan Kettering Cancer Center in New York as in-house test, and the FoundationOne assay that has been used for the Checkmate-227 trial and that has also been used in our study. However, both assays are not, as mentioned, based on fresh frozen samples but rather on FFPE tissue, as in our study too. (see Cheng D. et al doi: 10.1016/j.jmoldx.2014.12.006 for the description of the MSK-IMPACT panel as well as the associated FDA report: https://www.accessdata.fda.gov/cdrh_docs/reviews/DEN170058.pdf on page 2 in the sample preparation chapter. Information on the FoundationOne assay is given in Chalmers Z et al. https://doi.org/10.1186/s13073-017-0424-2 as well as again the associated FDA report: https://www.accessdata.fda.gov/cdrh_docs/pdf17/P170019C.pdf and the sample information sheet provided by Foundation Medicine: https://assets.ctfassets.net/vhribv12lmne/6ms7OiT5PaQgGiMWue2MAM/52d91048be64b72e73ffa0c1cab043c0/F1CDx_Specimen_Instructions.pdf).
Based on the data published using the two assays, which are based on several thousand tested patients (see Samstein et al. 2019), we are convinced that FFPE tissue is indeed suitable for TMB testing in routine clinical care.
Likewise, only one of the reported tests, MSK-IMPACT, sequences healthy patient tissue derived from donor blood to account for germline mutations, while this is not the case in the FoundationOne test or the Oncomine TML panel reported here. The MSK-IMPACT approach has certainly advantages as it allows filtering for each patient, however it is not considered to be a requirement for reliable genetic testing (see Sun JX et al. 2018 and Hiltemann S et al. 2015). Additionally, the FoundationOne assay has received FDA approval for genomic testing of treatment-related biomarkers across several genes without the need for a germline sequencing control, highlighting that the addition of a real germline sequencing control is certainly an advantage but not a requirement for precise variant identification using NGS.
The number of TMB (/Mbp) is required.
Reply: We agree with the reviewer that providing all the TMB is important and we have therefore added all the data for each patient as Supplementary data 2.
There is too much difference in number between lung cancer and malignant melanoma cases. In addition, there are some cases of lung cancer not receiving ICI treatment, and the variation of these cohorts is too large, which seems to be unsuitable for analysis.
Reply: We agree that two equally sized cohorts would have been advantageous. However, the primary goal of the study was to assess the correlation between the Oncomine TML panel and the FoundationOne assay in Lung Cancer patients as the Checkmate-227 trial was the only prospective TMB trial and it had been performed in lung cancer. The smaller Melanoma cohort was included to investigate whether we will see tumor-specific differences in the correlation between the Oncomine and the FoundationOne test (as discussed in the manuscript). We therefore do not have the same number of patients in the different groups.
Additionally, patient testing was requested when patients were reported to receive subsequent immunotherapy (intention-to-treat population). However, the initial decision could change sometimes due to previously unknown medical conditions or patient preferences as well as the initiation of palliative care rather than treatment with a curative intent. As also mentioned, some patients were subsequently treated at remote hospitals where we could not get access to their full clinical data. We therefore apologize that we were not able to provide the clinical data for all patients, however, those patients were still included for the assessment of correlation between the two assays.
We agree that the statistical power is consequently limited, and we have revised the discussion of the manuscript to better address this important limitation of our study.
Reviewer 3 Report
- In this work, Heeke et al. have studied the comparison between the OTML panel and Foundation One test to assess tumor mutational burden and to predict durable clinic benefit to immunotherapy (anti-PD1 drugs). The manuscript is well written but on my opinion is difficult to follow because the number of patients in each figure differs.
- Lane 74. In the introduction the authors have commented that OTML panel cover 1.65Mb of DNA, whereas in the supplemental information 1.7 Mb. Please modify.
- Supplementary Figure 1. Which are the differences between FMI failed and FMI TMB failed?
- Please provide the number of patients represented in Supplementary Figure 5.
- Supplementary Methods 1. Could you explain why samples were required to have a tumor size of at least 25 mm²?
- Supplementary Figure 6 is difficult to understand. Please indicate in the figure legend that the data correspond to 8 tumor samples.
- Figure 2. Please explain why the number of patients tested with NSCLC is 36 and in the melanoma cohort 24. Why in figure 3 the numbers of patients differ? NSCLC N=35 and in melanoma N=29.
-Supplementary Figure 9, 11 and 12. Why the number of patients represented is 24?
- Discussion. It would be interesting to discuss why the TMBhigh cut-off differ between OTML and FO panel in melanoma (5 vs 18 muts/mb) and why the cut-off in melanoma for OTML panel is lower than NSCLC.
Author Response
- In this work, Heeke et al. have studied the comparison between the OTML panel and Foundation One test to assess tumor mutational burden and to predict durable clinic benefit to immunotherapy (anti-PD1 drugs). The manuscript is well written but on my opinion is difficult to follow because the number of patients in each figure differs.
Reply: We apologize for the confusion caused by the different number of samples. We have done the evaluation for each cohort twice. First, we used all the available data to demonstrate the performance of each test individually. However, not all samples have been successfully assessed in all assays which consequently does not allow us to make direct comparisons (Figure 2B and Figure 5B). We have therefore included another analysis where only samples where all data for all the respective assays (TMB in FoundationOne, TMB in Oncomine TML, IHC for PD-L1 and CD8) were present. This would consequently allow the direct comparison of the different biomarkers as the same sample subset has been used for the analysis. However, we totally agree that this might be confusing for the reader and we have now only included samples where we have TMB for all samples as well as TMB obtained successfully from both samples (independent of the IHC results) (Figure 2C and Figure 5C). We have consequently updated Figure 2 and Figure 5 as well as the values in the text accordingly.
Additionally, we have clarified this in the discussion of the revised version of the manuscript and we have merged the ROC curves together (Figure 2B/C and Figure 5B/C) to make it easier understandable and to reduce the number of panels.
To also allow comparison of TMB with the biomarkers obtained from IHC assessment, we have included the TMB panels in supplementary figure 9.
- Lane 74. In the introduction the authors have commented that OTML panel cover 1.65Mb of DNA, whereas in the supplemental information 1.7 Mb. Please modify.
Reply: We apologize for the confusion. The only difference is the number of decimal places. We have now used 1.65 Mb throughout the manuscript.
- Supplementary Figure 1. Which are the differences between FMI failed and FMI TMB failed?
Reply: When receiving FoundationOne reports that had some problems during the testing, there are basically two possibilities: Either the whole sequencing run was not successful (“FMI failed”) or the mutations were called but the algorithm was not able to generate a reliable TMB value (the exact underlying problem was unfortunately not communicated to us; thus, we cannot give deeper explanations). Those samples are classified as “TMB failed” as only this part is missing. We have clarified this in the respective figure legend.
- Please provide the number of patients represented in Supplementary Figure 5.
Reply: We agree with the reviewer that information is important and we have provided it in the revised version of the manuscript.
- Supplementary Methods 1. Could you explain why samples were required to have a tumor size of at least 25 mm²?
Reply: The 25 mm² section size is required for testing using FoundationOne. We have now clarified this in the relevant section.
- Supplementary Figure 6 is difficult to understand. Please indicate in the figure legend that the data correspond to 8 tumor samples.
Reply: We apologize for the confusion regarding supplementary figure 6. We have now added the sample name and we have clarified the underlying data in the figure legend.
- Figure 2. Please explain why the number of patients tested with NSCLC is 36 and in the melanoma cohort 24. Why in figure 3 the numbers of patients differ? NSCLC N=35 and in melanoma N=29.
Reply: We agree with the reviewer and apologize for the error. We have now corrected the survival plot of the melanoma data in the revised version of the manuscript. However, this does not change the conclusion of the manuscript. Regarding the 35 patients in the NSCLC cohort: There was a small typographical mistake in the associated table which we have now corrected. This did not change the results.
Additionally, we have separated the Survival Analysis of Lung Cancer (now Figure 3) and Melanoma (now Figure 6) to make it more comprehensible for the reader.
-Supplementary Figure 9, 11 and 12. Why the number of patients represented is 24?
Reply: To allow the direct comparison of results, we have taken the samples for which all of the shown data was available in the same patients (a subset of 24 samples). We have updated the figure legends to clarify this and apologize for the confusion.
- Discussion. It would be interesting to discuss why the TMB high cut-off differ between OTML and FO panel in melanoma (5 vs 18 muts/mb) and why the cut-off in melanoma for OTML panel is lower than NSCLC.
Reply: We agree that the selection of the cut-off value is crucial for the analysis. Selection of cut-offs has been done in previous studies by selecting certain percentiles (for example 25th or 50th percentile) or calculating the cut-off using receiver operator characteristics (ROC). We considered the latter approach to be more reliable as it puts the selection on a statistical ground, free from human bias. We consequently calculated all the cut-off values using the minimal distance method and as this was reported to be very reliable (see Hajian-Tilaki K 2018, doi.org/10.1177/0962280216680383, Ilker U et al. 2017, doi: 10.1155/2017/3762651).
For lung cancer samples, where we see a good association of TMB with durable clinical benefit under immunotherapy, the minimal distance method and Youden index provides a good rational for the cut-off selection.
In the case of Melanoma, the discrepancy for the cut-off values is consequently based on the mathematical calculation. As the association with clinical response is very weak, the calculated cut-offs are relatively less meaningful and consequently chooses two very different cut-off values for the respective tests. This might not be clinically meaningful (and certainly is not in the provided dataset) but it is still based on the mathematical calculations. In a larger dataset with more statistical power, this cut-off might have been calculated differently and to clarify this we have now modified the manuscript accordingly.
Round 2
Reviewer 2 Report
In this revised manuscript, issues pointed out have been improved. However, there seems to remain some major limitations.
It should be clarified whether TMB predicted by AmpliSeq and Ion sequencer system is co rerated with TMB determined by whole exome capture sequence using Illumina sequencer system, because there seems to be a discrepancy between TMB with FO and OTML (Figure 4). How is driver mutational status in the lung cancer cohort? Does the number of driver mutations vary between TMB high and low? Is there multi collinearity between TMB and these driver gene alterations which may well be the predictor of ICIs?
Author Response
We would like to thank all the reviewers for their critical revision of the manuscript and the helpful comments that helped us to improve the quality of the manuscript. We would like to clarify some comments regarding the second revision of Reviewer 2 that we do not consider to be justified at this point now. Despite some comments that we have well addressed in the revised version of the manuscript, one of the main points was that the Reviewer did not consider the use of FFPE tissue to be possible or reliable for the assessment of tumor mutational burden. We have addressed this issue extensively in the manuscript and discussed the limitations of the use of FFPE tissue. However, we are convinced that our data demonstrates that FFPE tissue is useful for TMB assessment and our data is confirmed by the association of TMB with clinical response in NSCLC patients. Our results are here strongly in line with many recent high impact publications to which we refer in the rebuttal to the reviewer as well as in the discussion. Thus, the challenge to use TMB as a predictive biomarker in daily practice is to use FFPE tissue and certainly not frozen tissue material which is not available from a practical point of view in daily practice… We consequently disagree with the reviewer concerning this point and hope that this will not be considered in the final editorial decision. Additionally, we saw that in the second round of Revision, novel criticism, that was not mentioned in the first round of revision, appeared. We give some comments below, but this criticism needed to be address initially and not in the second round (so we are wondering what to expect for new criticisms in a potential third round, if any).
While there is still criticism regarding the use of FFPE tissue, we hope that we have well addressed in the first round of revision. However, the reviewer raises new concerns about the influence of driver mutations which could influence the response to immunotherapy independently of TMB status. While this is certainly an interesting question, we have to state that our study was not statistically powered to look for driver mutations, nor were there many driver mutations in the patient cohorts. We have one patient with an EGFR S768I mutation for which we could not find any studies looking at immunotherapy in the literature. Additionally, the patient showed durable clinical benefit which is contrary to EGFR L858R or exon 19 deletions which are usually reported to be associated with a non-response. Additionally, we have few patients with EGFR amplifications where literature concerning ICI treatment is also missing. Lastly, our results are in total agreement with the results published recently by Samstein et al (Nat. Genet. 2019) where driver mutations were not excluded for the analysis. Consequently and based on the low prevalence, we do not consider the presence of driver mutations to be a critical issue. Additionally, we cannot completely follow why this issue was not mentioned in the first round of the revision which would have allowed us to directly address this point in our rebuttal to the reviewer.
Reviewer 2
In this revised manuscript, issues pointed out have been improved.